# The microbiome of the marine flatworm *Macrostomum lignano* provides fitness advantages and exhibits circadian rhythmicity

Yuanyuan Ma[1], Jinru He [2], Michael Sieber[3], Jakob von Frieling[1], Iris Bruchhaus [4], John F. Baines[5,6], Ulf Bickmeyer [7] & Thomas Roeder [1,8]✉

The close association between animals and their associated microbiota is usually beneficial for both partners. Here, we used a simple marine model invertebrate, the flatworm *Macrostomum lignano*, to characterize the host-microbiota interaction in detail. This analysis revealed that the different developmental stages each harbor a specific microbiota. Studies with gnotobiotic animals clarified the physiological significance of the microbiota. While no fitness benefits were mediated by the microbiota when food was freely available, animals with microbiota showed significantly increased fitness with a reduced food supply. The microbiota of *M. lignano* shows circadian rhythmicity, affecting both the total bacterial load and the behavior of specific taxa. Moreover, the presence of the worm influences the composition of the bacterial consortia in the environment. In summary, the *Macrostomum*-microbiota system described here can serve as a general model for host-microbe interactions in marine invertebrates.

[1] Kiel University, Zoological Institute, Molecular Physiology, Kiel, Germany. [2] Kiel University, Zoological Institute, Cell and Developmental Biology, Kiel, Germany. [3] Max-Planck Institute for Evolutionary Biology, Dept. Evolutionary Theory, Plön, Germany. [4] Bernhard-Nocht Institute for Tropical Medicine, Hamburg, Germany. [5] Kiel University, Medical Faculty, Institute for Experimental Medicine, Kiel, Germany. [6] Max-Planck Institute for Evolutionary Biology, Group Evolutionary Medicine, Plön, Germany. [7] Alfred-Wegener-Institute, Biosciences, Ecological Chemistry, Bremerhaven, Germany. [8] German Center for Lung Research (DZL), Airway Research Center North, Kiel, Germany. ✉email: troeder@zoologie.uni-kiel.de

Almost all animals live together with associated microorganisms. From the perspective of host-associated microorganisms, whose entirety forms the so-called microbiota, the animal host provides a suitable ecological niche[1]. However, this link between the host and associated microbiota is crucial also for various life history traits of the host, with particular emphasis on behavior, development, and growth, as well as fecundity and lifespan[2–4]. More precisely, the natural microbiota often supports the development and metabolism of the host and thus offers an important fitness advantage. This also applies to humans where dysbiotic events of the microbiota are associated with several diseases[5,6]. Understanding this complex interplay between microbiota and host, and the effects of an out-of-balance interaction requires appropriate animal models[7,8]. The primary purpose of this is to enable a deeper, and more mechanistic, understanding of these complex associations[7,9,10]. In addition to the classical models of biomedical research, a few more animal models have been established that are very informative. In most cases, these are simply organized invertebrates with an equally simple microbiota, in which the most important bacterial representatives are not only cultivable but ideally also amenable to manipulation[9,10]. Here, the ability to generate gnotobiotic animals and the option to specifically manipulate the composition of the microbiota have provided insights into the proximate and ultimate processes underlying host-microbiota interactions[11–14].

These simple model systems can be broadly divided into two types of host-microbiota association. While one group of these models shows extremely simply structured interactions that are essential for the host, the second type of model is intended to represent the more standard case of a host-microbiota interaction. The first category comprises symbioses such as the Aphid/Buchnera system, very specific associations observed in different Hemiptera species[15–17], or wood-digesting insects, among which the termites stand out with their specific microbiota[18]. The study of these systems, which include bobtail squid/Vibrio fischeri interactions, has provided essential insights regarding those mechanisms underlying, e.g., the specificity and efficiency of colonization[19]. The second group includes, in particular, model systems that are used as examples of the standard form of host-microbiota interaction, in which the association is very important[20], but the composition of the microbial community is not fixed and is often subject to stochastic influences[21,22]. The classic models of biomedical research, mouse, zebrafish, Drosophila, and C. elegans are representatives of this type of host-microbiota interaction. For new models of this category, criteria should be met that have been formulated more recently; here, simple organization, low generation time, genetic manipulability, and relatively simply structured microbial communities are particularly noteworthy[9,10]. Besides this, the possibility to understand the mechanisms underlying the formation of the microbiome, the host-microbiota evolution, and the nature of the association are of great importance[23].

The need to establish new marine models for microbiome research appears somewhat counterintuitive, as several highly informative microbiome studies with marine models are available. However, it must be noted that these are usually very specialized systems in which the microbial partner is what makes successful host life possible in the first place. First and foremost, chemosynthetic symbioses have been characterized in detail[24,25]. In addition, two other systems were the focus of interest, (1) sponges, in which microbial symbionts play a central role for the holobiont, without which life is hardly possible[26–28], and (2) corals, in which the bacterial symbionts also perform highly specific tasks[29,30]. However, models are missing that pursue a lifestyle like that of the first bilaterian animals, being mobile and living in or on the seafloor[31,32], especially since this ecotype is so far not adequately represented. To fill this gap, we chose a representative that meets the requirements of new models to be established in host-microbiota research[9,10]. The plathelminth Macrostomum lignano was chosen because it is small, has a short generation time, is transparent, can be genetically manipulated, and is already used as a model organism in various research areas[33,34]. It has been successfully used in different fields of research including aging, development, regeneration, or stem cell biology[35–39]. The worm's genome assembly and annotation were available in 2015[40] and transcriptome assemblies showed stage- and organ-specific expression signatures[41]. Importantly, a series of mature experimental methods have been established, e.g., in situ hybridization[37], RNA interference[42], and transgenesis[43].

We analyzed the microbiomes of different developmental stages, but also of the environment, especially of the food they graze on. With this, we could identify candidate resident members of the microbiota and showed daily cycling of the abundances of the microbiota as well as of some bacterial members of the microbiota. Moreover, using gnotobiotic animals we were able to show that the microbiota does not influence lifespan directly, but enhances the fitness of the host in times of food restriction.

## Results

**Microbiome composition of M. lignano**. To provide the basis for further studies, we analyzed the microbial consortia of different developmental stages of the flatworm (Macrostomum lignano) and the relevant environment. In detail, we studied the microbial composition of the following sources: (1) The algae (Nitszchia curvilineata) with the medium before contact with worms (denoted "Algae"), (2) the medium after 3 weeks of colonization with worms, but after removal of them (denoted "pre-conditioned medium", PCM), (3) eggs laid by the mature worms (denoted as "Eggs"), (4) worms collected 1–2 days after hatching (denoted "immature worms", IMWs), and (5) worms cultured on the algae for at least 3 weeks (denoted "mature worms", MWs). The bacterial taxa were defined for all samples via the V1-V2 region of the bacterial 16S rRNA gene.

We found diverse microbiota associated with the different developmental stages of M. lignano kept under these conditions. In our studies, 4120 amplicon sequence variants (ASVs) were identified (Supplementary Data 1) and the top 37 ASVs, which represented over 99% of the total bacterial abundance were selected for analysis of composition at the family level (Fig. 1). Preceding the more detailed description the ASV are mentioned, which were found in the different stages. Algae, ASV1-8, ASV10-25 and ASV33, PCM, ASV1-25, ASV31, ASV33, ASV37; Eggs, ASV1-29, ASV31, ASV34-37; IMWs, ASV1-15, ASV17-18, ASV23, ASV25, ASV27, ASV29, ASV31, ASV34-35, and ASV37-38; MWs, ASV1-6, ASV8-17, ASV 19-10, ASV23, ASV25, ASV27, ASV29, ASV 30-35, ASV37-38. Algae, without any contact with worms, showed a specific microbial community compared to all other samples that had contact with worms (Fig. 1A). Here, Stappiaceae, Rhizobiacea, and Hyphomonadaceae showed the highest relative abundances (Fig. 1, Algae vs. PCM, $p < 0.0001$). However, the fact that worms lived for a certain time on this medium substantially and significantly changed the bacterial community, with Terasakiellaceae, Rhodobacteracea, Cyclobacteriaceae, and unclassified bacterial families found in higher concentrations in all tested PCM samples (Fig. 1A, Algae vs. PCM, $p < 0.0001$). This difference is also evident in worm samples. The surface-washed eggs contain 6 main bacteria groups, including Rhodobacteraceae, Rhizobiaceae, Phycisphaeraceae, Hyphomonadaceae, Alteromonadaceae, and unclassified bacterial groups. This bacterial composition was apparently

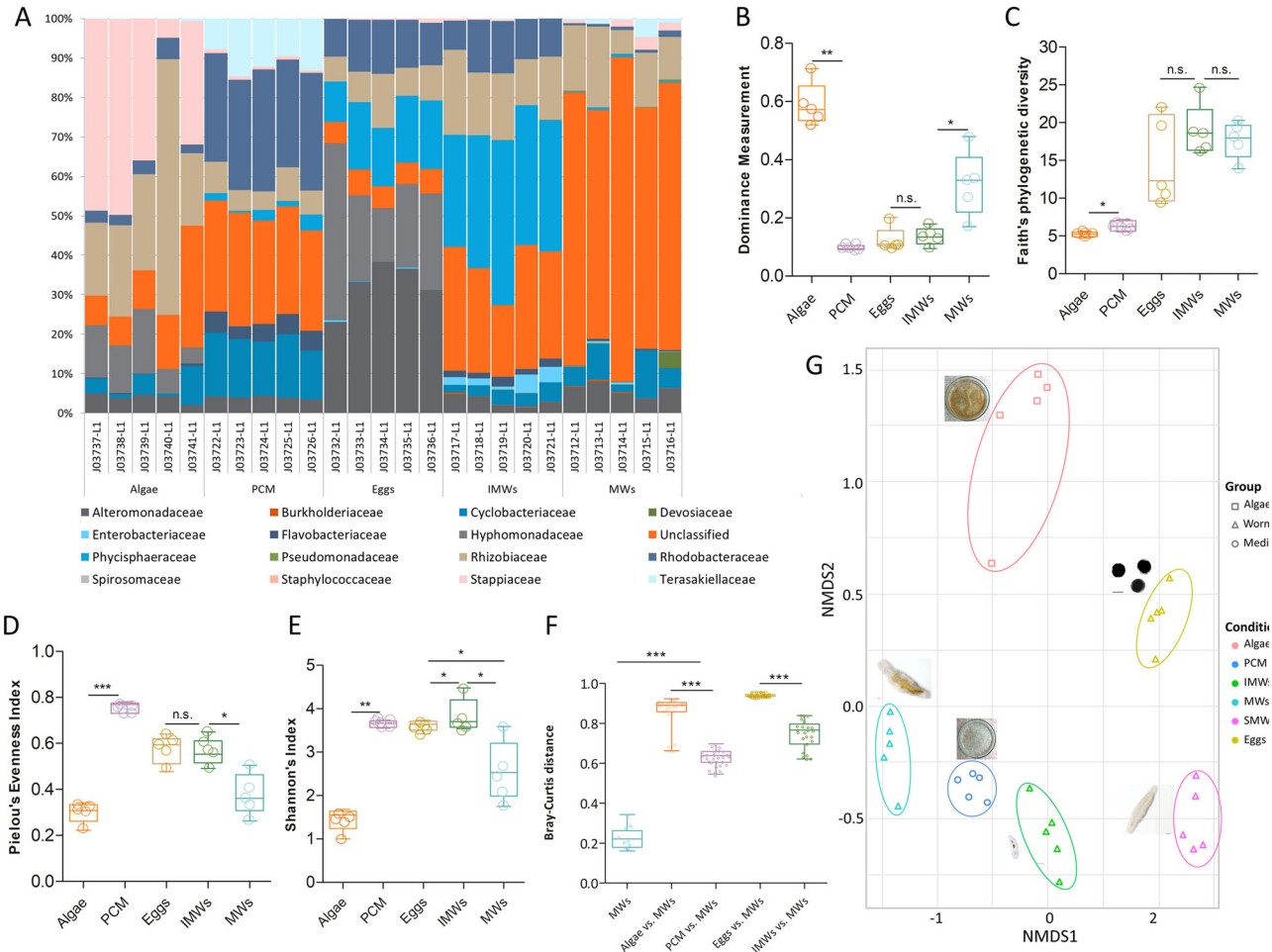

**Fig. 1 Bacterial composition in samples taken from algae, PCM, and different developmental *M. lignano* stages.** In **A**, relative bacterial abundances of algae, pre-conditioned medium (PCM), eggs, immature and mature worms (IMWs and MWs). Colored rectangles are used to distinguish different families of microorganisms. In **B–E**, Alpha-diversity metrics, including the Dominance measure, Faith's phylogenetic diversity, Pielou's evenness, and Shannon's index show the diversity of bacterial samples from different sources. In **F**, Bray-Curtis distances of all samples in comparison with MWs are shown. **G** The non-metric multidimensional scaling (NMDS) plot based on the Bray-Curtis distances analyses (stress value = 0.069). The different microbial colonizer taxa are listed with color-coded assignments. Each group employed 5 mature or 10 immature animals with five biological replicates. Asterisks denote significant differences between samples (one-way ANOVA with Turkey Post-hoc test), $*p < 0.05$, $**p < 0.01$, $***p < 0.001$.

different from that of the parental mature worms, where unclassified bacteria account for roughly 60% (Fig. 1A, Eggs vs. MWs, $p < 0.0001$). While comparing IMWs with MWs, it became apparent that Rhodobacteriaceae, Phycisphaeraceae, and Enterobacteriaceae were significantly increased in all examined IMWs groups, while unclassified bacteria families were reduced to ∼ 30% (Fig. 1A).

Estimating the α-diversity metrics (Dominance measure (Fig. 1B), Faith's phylogenetic diversity (Fig. 1C), Pielou's evenness (Fig. 1D), and Shannon's index (Fig. 1E)) were estimated using QIIME 2 after samples were rarefied to 23152 sequences per sample. Here, the dominance index (Fig. 1B) was by far the highest in the algae samples, indicating lower diversity compared to those samples derived from or in contact with worms. These differences between der algae and the other samples are also confirmed by the calculations of the Faith index (Fig. 1C), Pielou's evenness (Fig. 1D), and Shannon's index (Fig. 1E). Moreover, pre-colonization medium and early developmental stage animals (eggs and immature worms) have higher bacterial diversities than adult animals and algae samples (Fig. 1B–E). β-diversity evaluation by Bray-Curtis dissimilarity analysis clearly showed higher diversity in all samples associated

with the presence of worms (Fig. 1F). Displaying the dissimilarities as an NMDS plot with two dimensions revealed that all replicates of a peculiar sample type cluster together. All worm-associated samples show the greatest distances to the algae samples (Fig. 1G). The first and second axis separate all worms from algae and pre-conditioned medium (PCM) samples, indicating that the microbial community of the worm is distinct from its environment (Fig. 1G, $p \leq 0.0003$). Eggs, IMWs, and MWs all show separate clusters (Fig. 1G, $p < 0.001$). These results were supported by a Bray-Curtis PCoA distance analysis (Supplementary Fig. 1).

To analyze the composition of the microbiota and identify bacterial taxa that differ from the expectations of the neutral model in terms of abundance and frequency of occurrence, we tested the data of the algae and mature worm (MW) samples against the null hypothesis of a neutral model (Fig. 2)[22]. The bacterial community isolated from the algal samples followed mostly the expectations of the neutral model (Fig. 2A, Supplementary Data 2). The situation is more complex for the community derived from the MWs. Most bacteria were found according to these expectations indicating a stochastic type of colonization, but some taxa significantly differed from this

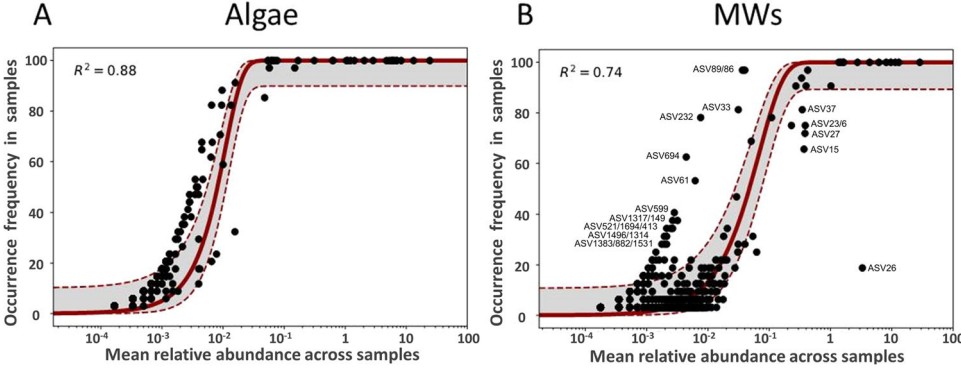

**Fig. 2 Relationship between mean bacterial taxa abundance across all samples and their frequency in algal and mature worm samples (MWs).**
Individual bacterial taxa are represented by dots. The solid line represents the neutral community expectations with the 95% confidence interval (gray area bordered by the dotted lines). In **A** the comparison of the algal samples and in **B** the comparison of the mature worm-derived samples is given. Points found to the left of the confidence interval represent bacterial taxa that are found in more samples than expected, but with lower abundances, whereas those found to the right of the confidence interval represent bacterial taxa that are found in fewer samples than expected, but there at high abundances.

(Fig. 2B). Here, taxa such as ASV232, 33, 89, 694, or 61 are found in more samples than expected but with lower abundances, whereas especially ASV26 (*Serratia marcescens*) is found in very high abundances in very few samples. Assignment of all ASV to bacterial taxa can be found in Supplementary Data 1. To evaluate whether these deviations from the expectations of the neutral model also occur in a similar way in the other samples containing *M. lignano*, we additionally tested the colonization pattern of immature worms (IMWs) and the pre-conditioned medium (PCM) samples. In addition to some bacterial species that cannot be correctly assigned to, we found 21, 30, and 7 groups of bacterial taxa, where the observed abundances were not compatible with the expectations of the neutral model in IMWs, MWs, and PCM samples, respectively. Supplementary Data 3 summarizes these taxa.

To elucidate if the microbiota of *M. lignano* also contains residential bacteria, we tested the bacterial composition of worms that were starved for 5 days and thus unable to replenish the microbial community by feeding (named SMWs). In the SMWs the amount of unclassified bacterial families was reduced, while increased relative amounts of Rhodobacteraceae, Rhizobiaceae, Phycisphaeraceae, and Alteromonadaceae were seen compared to MWs (Fig. 3A, SWMs vs. MWs, *p* < 0.0001). Interestingly, some bacterial families that are found in MWs were hardly found in SWMs, including Stappiaceae, Terasakiellaceae, and Rhodobacteraceae. As expected, starved animals showed a reduced total bacterial load if compared with non-starved ones (Fig. 3B, MW vs. SMWs, *p* < 0.0005). Comparing the α-diversities based on the Shannon index showed no significant differences between MWs and SMWs (Fig. 3C). β-diversity based on Bray-Curtis dissimilarity showed higher bacterial diversity in SMWs compared to MWs (Fig. 3D). Displaying the dissimilarities as a two-dimensional NMDS plot showed that the respective replicates cluster together and that they can be easily separated by the first dimension (Fig. 3E).

**Analyzing the relevance of the microbiota – comparing germ-free (GF) and re-colonization (RC) animals.** To infer the relevance of the native microbiota of *M. lignano*, we generated germ-free (GF) animals through antibiotic treatment. To allow a direct comparison between germ-free (GF) and conventional worms, and to exclude the antibiotic treatment as a confounder, all animals were first made germ-free before one group was recolonized with the native microbiota and the other group remained sterile. The outline of the procedure is shown in Fig. 4A. For the worms of the GF group, no bacterial colonization was seen after plating

homogenized worms on marine-agar plates and almost no bacteria were detected through amplification with the universal bacterial primers (Fig. 4B). Both germ-free and re-colonized animals can live about 50 days without food supply, which shows that the presence of a native microbiota had no effect on the lifespan under these conditions (Fig. 4C).

We found that the presence of microbiota had no significant effect on the worm's progress of development. All experimental juveniles grown either in sterile (SM) or re-colonization medium (RCM), developed into adults after about 6–7 days. To quantify the effects on fitness, meaning offspring production, we measured the number of offspring produced by cohorts of 10 mature worms over a period of 16 days under standard laboratory conditions (20 °C; Fig. 4D). Constantly refreshing the medium ensures an optimal supply with nutrients. Under these conditions, almost identical numbers of offspring were generated in GF and RC worms (Fig. 4D, left). Reducing the food availability by a different nutritional regimen where food was replenished every third day led to slight starvation, and under these conditions, the GF worms showed a reduction in offspring production by about 35% compared with the recolonized worms subjected to the same regimen (Fig. 4D, right, *p* < 0.001). The difference between germ-free and recolonized worms was highly significant (Fig. 4B, right). The recolonized worms subjected to this reduced nutritional supply showed an offspring production that was almost identical to that under ad libitum conditions. We subsequently tested the feces output of the mature worms. They usually produce two different types of feces, rod-shaped dark fecal spots (2, Fig. 4E, inset) and smaller, round, and oblique spots (1, Fig. 4E, inset). RC animals produced far more type 2 spots than type 1 spots (Fig. 4E). For GF animals the ratio was exactly the opposite, with more than 90% being type 1 spots. These differences between GF and RC worms were highly significant (*p* < 0.001 each).

**The microbiota of *M. lignano* shows a circadian rhythmicity.** To evaluate whether the worm microbiota and/or its individual members show circadian rhythmicity, we kept the animals for two weeks in a climate chamber with a 12 h/12 h day/night rhythm. Thereafter, worms were isolated and analyzed every 3 h within a 24-h period (following the Zeitgeber time). These samples were used to evaluate the relative and total bacterial abundances of different bacterial orders in the MWs and the PCM samples, respectively. We performed a neighbor-joining (NJ) analysis with the top 50 ASVs, which covered over 99% of the detected bacteria representing 21 orders (Fig. 5). With this analysis, differences between the PCM (left side) and the MW (right

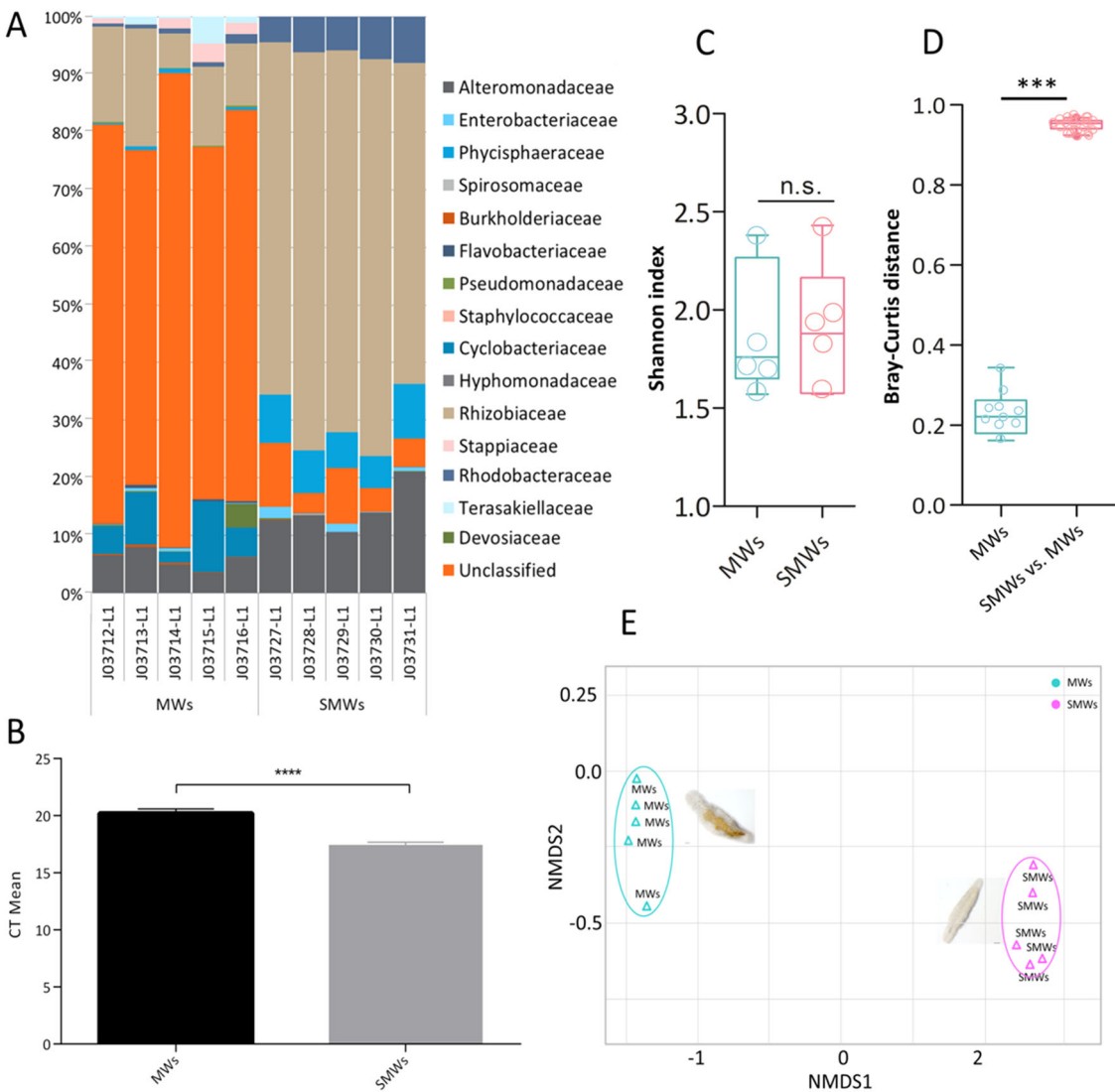

**Fig. 3 Diversity and bacterial load of mature worms (MWs) and starved mature worms (SMWs).** In **A**, **B**, the relative bacterial abundances and bacterial load of MWs and SMWs are shown. **C**, **D** Shannon index, and Bray-Curtis distances analysis show the diversity of the microbial community of bacterial samples from MWs and SMWs. **E** The non-metric multidimensional scaling (NMDS) plot based on the Bray-Curtis distances analyses. Each group employed 5 mature animals with five biological replicates. Asterisks denote significant differences between samples (t-test), ***$p < 0.001$, ****$p < 0.0001$, and n.s. no significant difference.

side) communities became apparent. Some bacteria were only apparent in the environment and never or rarely observed in the worm samples. This group comprises Alteromonadales, Oceanospirillales, and Caulobacterales, whereas other orders were specifically found in worm samples, e.g., some orders of the beta-Proteobacteriales and Flavobacteriales (Fig. 5). Moreover, the time-resolved analysis revealed substantial changes during the day both, in the PCM and the MW communities (Fig. 5). Here, the Rickettsiales showed the most interesting phenotype as they accumulated only at later time points of the day, which might indicate that they strictly depend on the circadian rhythms of the host (Fig. 5). Again, the identities of the different taxa could be found in the Supplementary Data 1. Based on these results, we focused on the circadian rhythmicity of the entire bacterial community as well as of selected bacterial groups (Fig. 6). The material from isolated worms was used to quantify the bacterial load via qPCR (Fig. 6A). According to these quantifications, the microbiota of *M. lignano* showed a clear circadian rhythm in the observation period (Fig. 6A). The lowest bacterial load was detected at Zeitgeber time 18 h, where the values were ~2.2-fold

lower than at the maximum at 12 h (Fig. 6A and Supplementary Fig. 2). The worms' microbiota is characterized by a relatively low diversity of the bacterial community when compared to the culture medium (PCM) at all time points analyzed (Fig. 6B). The lowest bacterial diversity of the worms was detected at Zeitgeber time 6 h based on the Shannon index (Fig. 6B). Peregrinibacteria (ASV3; Fig. 6C) and *Pelomonas* (ASV29; Fig. 6D) were selected as representatives to show this difference. Peregrinibacteria were mainly found in the PCM, while *Pelomonas* was at least 100-fold more abundant in the worm samples (Fig. 6C, D). Furthermore, we also analyzed if the abundances of these bacteria show circadian rhythmicity. Using the JTK program package, this cycling was obvious for the Peregrinibacteria found in the worm, but also for those found in the medium. Interestingly, this cycling showed a nearly mirror-image distribution of minima and maxima (Fig. 6C). In contrast, no cycling was observed in the worm and environmental samples for *Pelomonas* (Fig. 6D).

Bray-Curtis and Weighted Unifrac distance analysis separated the microbial communities of worms from those of their living substrates at every time point (Fig. 6E, F and Supplementary

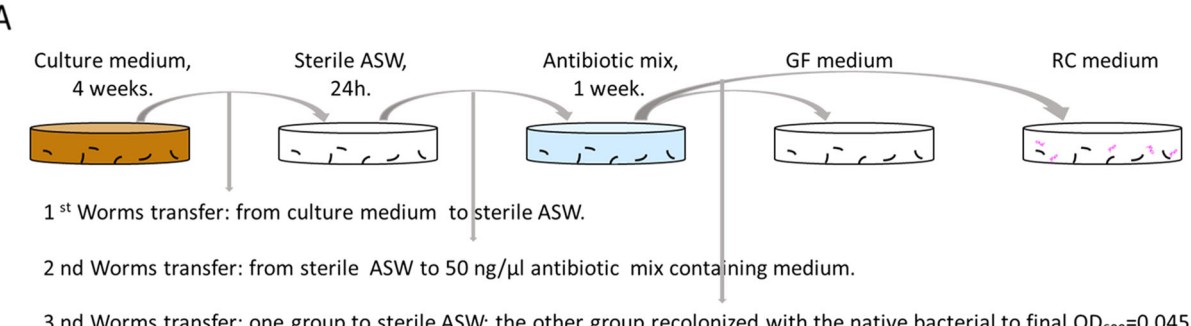

**Fig. 4 Survival, fitness, and fecal output of germ free (GF) and recolonized (RC) worms. A** The workflow to obtain germ free and recolonized animals. In **B**, the bacterial colonization and PCR results of GF and RC animals are shown. **C** The survival curves of GF and RC animals. In **D**, the population size after 16 days was quantified, starting with 10 mature worms under standard laboratory conditions (20 °C). Numbers of offspring of GF and RC animals under optimal food conditions (left) and under reduced food availability (right) are shown. In **E**, fecal spots were quantified in RC animals and GF animals. Feces output was classified into two categories: (1) small, half-baked, and transparent clumps; (2) integrated and brunet clumps. Vertical arrows are used to indicate the worm transfer process. Unpaired two-tailed Student's *t*-test. ****p* < 0.001; n.s. denotes no significant differences with Log-rank (Mantel-Cox) test. *N* = 15.

Fig. 3). The changes in the community observed in the PCM changed over time, where especially the samples Zeitgeber time 6 h and 9 h are completely apart from those of the other samples. Very similar patterns were observed for the corresponding worm samples (Fig. 6E, F).

## Discussion

An important goal of this work was to develop a marine model for microbiome research that could provide novel insights into

the interactions between host and colonizing microbiota. We chose *Macrostomum lignano*, a marine invertebrate, that, on the one hand, maintains a lifestyle typical of the earliest representatives of the bilaterians and, on the other hand, meets the criteria for new models in microbiome research[9,10]. The platyhelminth *Macrostomum lignano* shows exactly this lifestyle of the earliest Bilateria; it is a small and mobile marine invertebrate that moves on or in the sea ground and finds its food there[31,32]. It was easy to adapt it to the needs of microbiota studies because *M. lignano* is already an established model in other research areas[34,39,40,43]. In

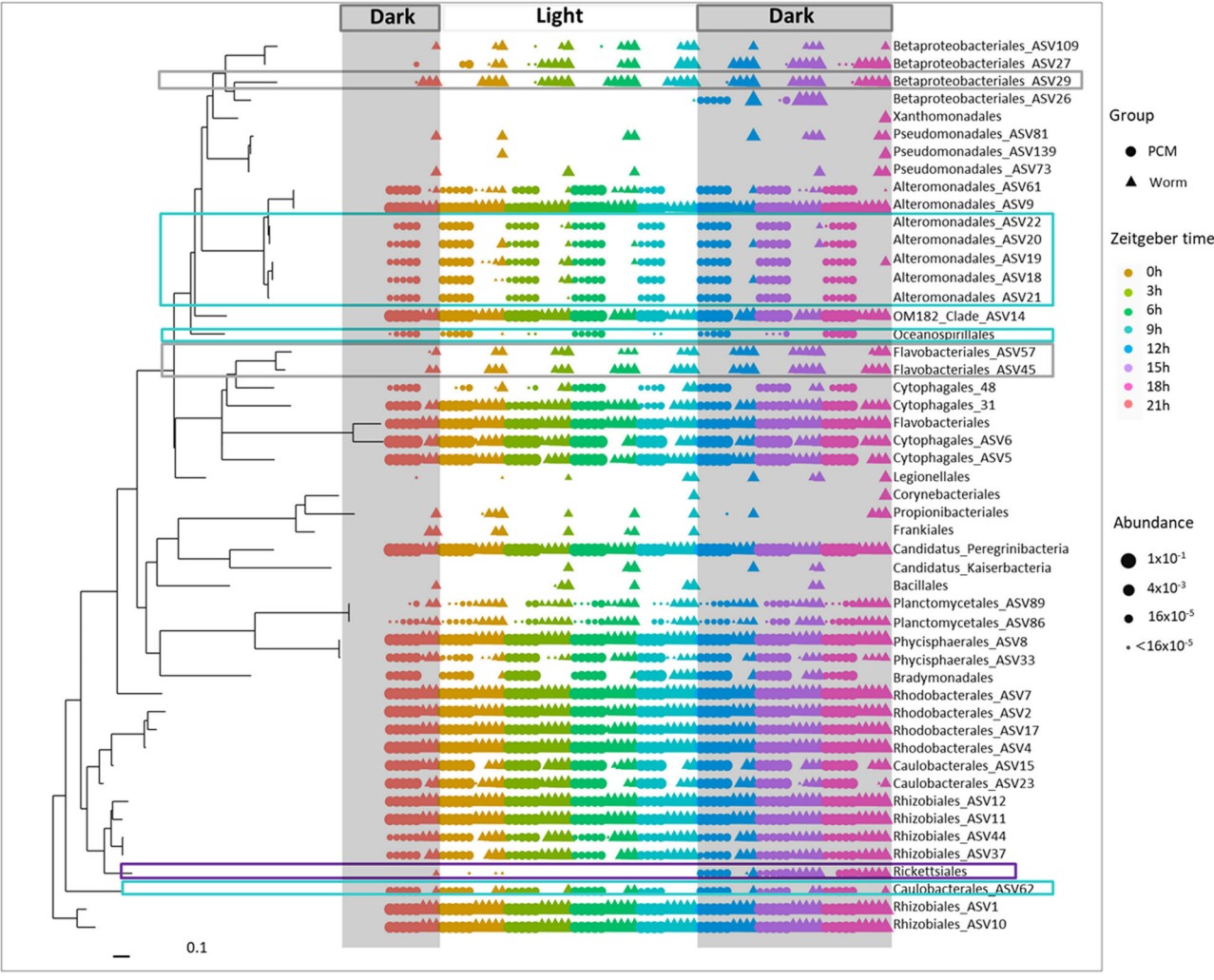

**Fig. 5 The phylogenetic relationships and bacterial abundances in both, worms, and their living substrates (PCM) during a 24 h period.** The top 50 ASVs were used for phylogenetic reconstruction[82]. The round symbols show the relative bacterial abundances at the given time point in the PCM samples, and the triangle symbols show the relative bacterial abundances in the worm samples. The chosen time points during the 24 h observation time were highlighted by the color code (expressed as Zeitgeber time), $N = 3–5$.

addition, since the bacteria colonizing it are marine environmental bacteria and these can often be cultured[44], and since we were able to show here that gnotobiotic animals can be generated, maintained, and quantitatively analyzed, *M. lignano* is well suited for microbiome studies. In the present work, we have characterized major features of the host-microbiota interaction and thus laid the groundwork for further studies. We found that (i) the different developmental stages of the flatworm each have specific microbiota, (ii) that these differ significantly from the environment, (iii) that the worm alters the microbial community of its environment, (iv) that the microbiota provides a fitness advantage for the host, and (v) that parts of the microbiota show a distinct circadian rhythm.

One of the key questions we addressed was whether the microbiota confers a fitness advantage for flatworms. Here, using gnotobiotic animals is a straightforward approach, as it allows direct attribution of the observed phenotypes to the presence or absence of a microbiota[9,45]. However, to avoid confounding effects caused by additional influences of the antibiotic treatment, it is recommended to compare only gnotobiotic with recolonized animals[46]. Comparison of the recolonized and germ-free animals revealed several similarities, which included life span without food intake on the one hand, and offspring production when food

was abundantly available on the other. Interestingly, we found a fitness advantage under restricted conditions. Here, the food intake was reduced, and the germ-free animals showed a significantly reduced fitness, whereas the recolonized animals showed a fitness exactly corresponding to that of animals kept under optimal feeding conditions. Such a situation, in which an improved response to food stress was observed, was also demonstrated in other models such as *Drosophila*[47]. In general, it can be argued that this host-microbiota interaction can strongly influence the ecological niche of the host and in many cases leads to a widening of this niche, which opens up new colonization opportunities as experiencing periods of limited access to food tend to be the normal case for most organisms[48].

A first result of this study was that all host (*Macrostomum lignano*) associated samples show a profile that is uniquely distinguishable from the environmental samples. All these communities differed significantly from those found in the environment, but also from those found in the conditioned medium. A picture like this, where the composition of the microbiota changes between developmental stages, appears to be seen for most organisms and may reflect the colonization history or the different physiological states of the corresponding host developmental stages[49–51]. Although the microbiota has been

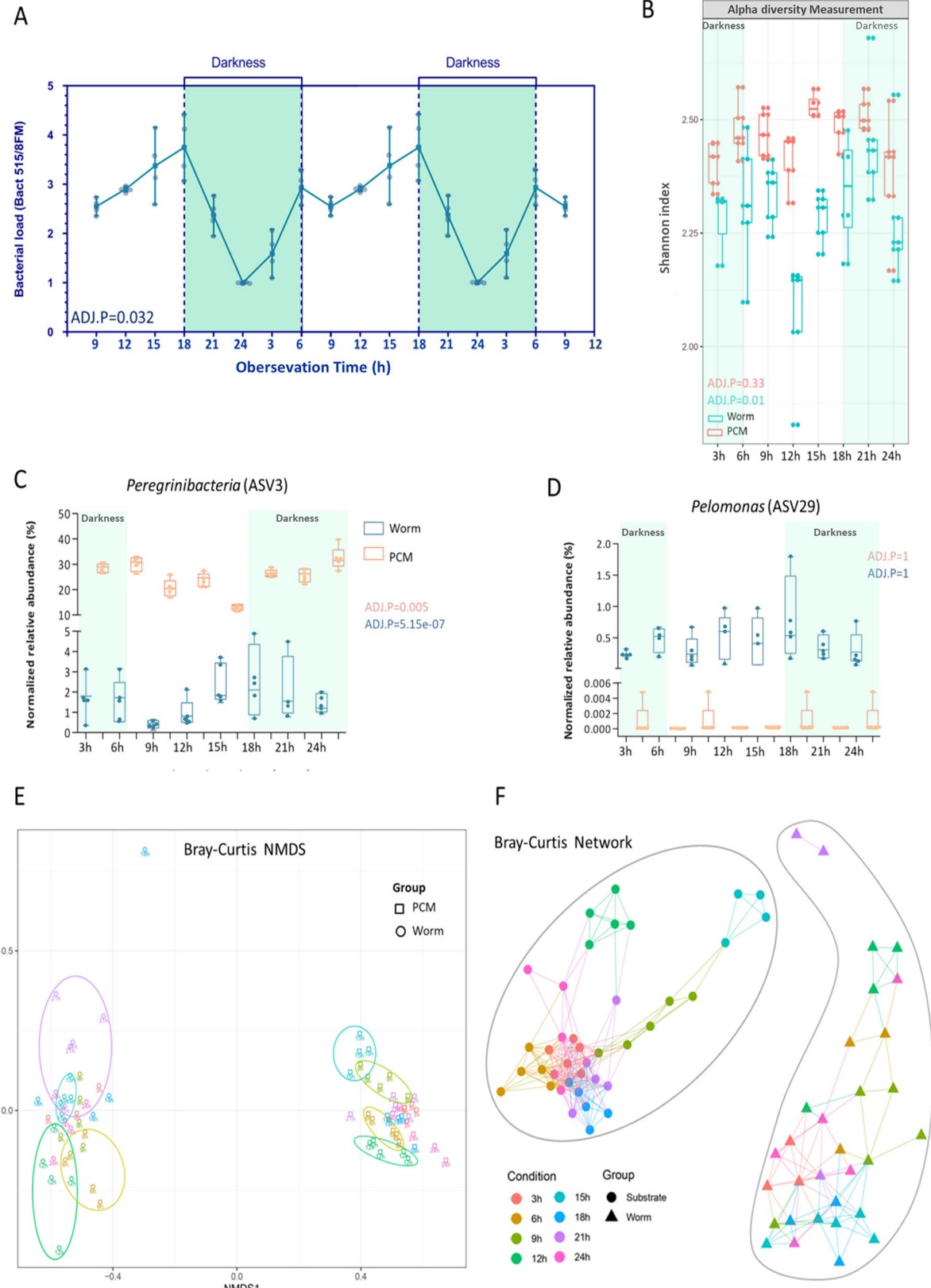

**Fig. 6 Microbial circadian rhythmicity of *M. lignano*, and composition characteristics of the microbiota of worms and PCM in the observation time points.** In **A**, the relative bacterial load is shown during the 24-h observation period (expressed as Zeitgeber time). In **B**, beta-diversity measurement of these bacterial samples is based on Shannon index analysis. **C**, **D** The relative abundances of the *Peregrinibacteria* and *Pelomonas*. **E**, **F** Bray-Curtis nonmetric multidimensional scaling (NMDS) and a network of worms and PCM in eight record time points. The Ellipsoids represent a confidence interval surrounding each group, stress value = 0.065. The color code indicates the time points in both **E** and **F**. The cycling statistical JTK outputs permutation-based p-values (ADJ.P) were defined. *N* = 5.

acquired during development from the environment, host-related factors, such as the host's immune system, shape this community during all phases of development[52]. Moreover, abiotic parameters such as pH may also contribute to this process. We observed that the intestinal tract of the worms is slightly acidic ($5.2 \leq pH < 6.8$), while the sea water is neutral to alkaline ($7.4 < pH \leq 9.0$; Supplementary Fig. 4). A pH dependency of the microbial composition has also been shown in other systems[53]. To distinguish residents and transient inhabitants, we have adopted a simple experimental approach. We let the worms starve for a few days and then looked at the remaining microbiota. This revealed a considerable shift in the composition of the community. On the one hand, there was a significant reduction of bacterial abundance, but at the same time, there was also a very significant increase in certain bacterial taxa. These include Rhodobacteraceae, Rhizobiaceae, Phycisphaeraceae, and Alteromonadaceae, which can therefore be considered as more resident parts of the microbiota. This distribution found here, in which some resident taxa and a larger number of non-resident taxa, so-called 'travelers', make up the microbiota, seems to represent the normal situation[54]. On the other hand, in some systems, no residents were found[55]. The specificity of colonization, especially of the most important life stage of *M. lignano*, the adult worms, is also supported by modeling colonization using the neutral model approach. Here, most bacterial taxa followed the predictions of the neutral model, but some showed deviations from this distribution. Of note is the high abundance of *Serratia* in very few samples indicative of a dysbiotic situation.

In addition to the role that the microbiota plays in fitness described in this work, the potential importance of the microbiota in regeneration remains to be discussed. This is mainly because plathelminths, including *M. lignano* are well-established models in the field of tissue regeneration[33,35,56]. Although we have not conducted experiments in this regard, it is reasonable to expect that the microbiota exerts a decisive influence on regenerative processes. In addition to the multitude of models in which the microbiota has been shown to be involved in regeneration processes[57], this question has already been addressed in two studies with planarians. In *Schmidtea mediterranea* bacterial infection and dysbiosis induced by tissue lesioning impaired the regeneration process[58]. In *Dugesia japonica*, indole production by a microbial colonizer also impaired regeneration[59]. Looking at the role of the microbiota in *M. lignano* in future studies might further unravel its general role in regenerative processes.

Another finding of this work was unexpected. Substantial differences were found between the bacterial consortia of the environmental samples and those of the conditioned medium (PCM). Although both share the same initial situation, the presence of the worms seems to lead to a substantial remodeling of the bacterial consortia in their environment. This means that not only the environment influences the microbiota of the flatworm, but also the holobiont consisting of *M. lignano* and its microbiota has a clear and long-lasting influence on the microbial composition of its environment. This aspect has hardly been considered so far but could turn out to be a means to enable true co-evolution between host and microbiota[60]. Similar types of influences have also been reported for fruit flies, where the presence of flies had a dramatic impact on the microbial consortia on the substrate[61]. To correlate the changes observed in the composition of the microbiota with specific traits would be highly desirable, but currently, we can only use information from other systems for discussion. *Oceanospirillales*, which was an abundant colonizer showing circadian rhythmicity were found at sites with salt intrusion[62]. For example, the composition of the intestinal medaka-microbiome was shown to change after osmotic stress and taxa such as *Alteromonadales* or *Rhodobacterales* were less

abundant[63]. Rhizobiaceae, which are among the candidates for persistent colonizers in *M. lignano*, are also found in other marine invertebrate microbiomes[64]. Phycisphaeraceae and Rhodobacteraceae are also taxa that were found in this study as being residents and have recently been shown to be part of the microbiome of the microalgae *Gambierdiscus*, where they might be relevant for the metabolic abilities of the holobiont[65].

Furthermore, we showed that the composition of the *M. lignano* microbiota is subject to a pronounced circadian rhythm. This applies to both the bacterial abundance and the relative composition of the microbiota. The differences between minima and maxima are more than threefold in terms of bacterial abundance, indicating that the influence of the microbiota on the host and vice versa change significantly during the day. Moreover, the composition of the microbiota was clearly distinct between different times of the day. These differences were not congruent with the differences in abundances that were maximal between the time points 18 h and 24 h, whereas maximal differences in terms of the microbial composition were seen between the time points 12 h and 21 h. The latter two time points had very similar abundances but were either on an ascending (12 h)- or a descending curve (21 h). Circadian rhythmicity of the microbiota has already been shown in holobionts from different animal groups. Of course, mammals are at the center of interest[66,67], but such a phenomenon could also be shown for arthropods[68]. Furthermore, circadian rhythms of microbiota could already be demonstrated in some marine invertebrate holobionts. This applies to Cnidaria, but also to molluscs, and sponges[69–72]. Mutual interactions between microbiota and host have been shown in different systems. In mammals, not only the influence of the host on the microbiota but also an inverse direction of this flow of control has been demonstrated[73,74]. In invertebrates, mechanistic information in this regard is much scarcer. The exception is the bobtail squid, of course. Here, the host mechanisms that contribute decisively to the circadian regulation of the abundance of the symbiont *Vibrio fischeri* were elucidated[19,70,75]. Despite these efforts, we are currently just beginning to understand this complex interplay and especially the underlying molecular mechanisms.

Taken together, we characterized the microbiota of the marine flatworm *Macrostomum lignano*. Most importantly, we were able to demonstrate a fitness advantage of a microbiota under conditions of reduced nutrient availability. Moreover, the different developmental stages of the flatworm show different microbiota, each of them being clearly different from the microbial consortium of the environment. We also found that the host, together with its microbes, can shape the microbial community in its environment. A small group of bacteria was identified as candidate resident members of the microbiota. Finally, we could identify a robust circadian rhythm of the microbial abundance, but also of the composition of the microbial community of *M. lignano*, pointing to a complex interplay between the host and its associated microbiota. In summary, we think that this work has laid the basis for follow-up studies in which this very interesting host-microbiota system will provide deeper insights into different aspects of this complex interplay.

## Methods

**Marine-agar plates**. To make the marine-agar plates, the following procedure was used: 10 g of tryptone, 1 g of KCl, 4 g of $MgCl_2 \cdot 6H_2O$, 10 g of NaCl, and 15 g of agar were filled up with 1 L of deionized water. The solution was stirred thoroughly and adjusted to pH 7.5. After autoclaving and cooling the liquid was poured into 60 mm sterile petri dishes under the clean bench. After they solidified at room temperature (RT), the plates were stored at 4 °C before use.

**Animal culturing**. Animal culturing was essentially done as described earlier[76]. *Macrostomum lignano* (DV1 line) was originally collected from sediments of the

Adriatic Sea and obtained from the Department of Zoology and Limnology, University of Innsbruck, Austria, and then reared in petri dishes, fed with *Nitszchia curvilineata* and cultured in Guillard's F/2 medium (Sigma G0154). Both animal and diatom cultures were incubated at RT (20 °C). 3–4 weeks old adults and the appropriate developmental stages were used for the current studies.

**Antibiotic treatment for both algae and worms.** Antibiotics (Anti) containing sterile Guillard's f/2 medium included 50 µg/ml ampicillin, 50 µg/ml streptomycin, 50 µg/ml neomycin, 50 µg/ml rifampicin and 50 µg/ml spectinomycin. The algae medium was treated with Anti Guillard's f/2 for 4 weeks and the medium was replaced with fresh medium at 2-day intervals. The newly hatched juveniles were washed in the sterile Guillard's f/2 medium for 2 days. Then the animals were transferred into the sterile algae medium for 4 weeks until the animals were grown up. The medium was renewed every 2–3 days. The germ free (GF) animals were cultured in sterile Guillard's f/2 medium for 2 days to get rid of potential antibiotic effects before measuring the lifespan. We used the specific 16s rRNA gene primers (forward primer 27 F (AGAGTTTGATCMTGGCTCAG)/reverse primer 338 R (TGCTGCCTCCCGTAGGAGT)) for PCR and only a very weak band was detected in the PCR product of GF animals.

**Recolonization.** For recolonization, the 3–4 weeks worm cultured medium was filtered through a 0.8 µm filter and adjust with sterile Guillard's f/2 medium to the final $OD_{600}$ of 0.045. It was the normal bacterial concentrations under lab conditions. For the GF group, a 0.2 µm filter was used and the final $OD_{600}$ was 0. Worms were grown in the 24-well plates that each well was filled with 1 ml of filtered homogenates. The GF worms recolonized with bacterium were named as "recolonization" (RC) worms.

**Worm lifespan.** In total fifty worms (10 animals per replicate) were used in the experiment of the lifespan. To lower the risk of contamination and other interfering factors, e.g., evaporation, the medium was changed every day.

**Feces output measurement.** To slow feces decay during the observation time (24 h), we employed 3 animals per group (5 replicates). All animals were cultured in 24-well plates with a sterile medium for 10 days. The medium was replaced every day. Reducing the number of animals in this assay made the quantification of fecal spots easier and more reliable.

**Gut pH staining.** The pH measurements were performed by using the indicator dyes, m-cresol purple (657890, Merck, Sigma-Aldrich, Steinheim, Germany), and bromo-cresol purple (B5880, Merck, Sigma-Aldrich, Steinheim, Germany). Animals were stained with these 2 dyes using 0.1% final concentration at a ratio of 1:1 for 4 h. Before taking photographs, they were anesthetized with 7.14% $MgCl_2$ for 20 min.

**M. lignano and substrate DNA extraction for 16 S rRNA gene sequencing.** The worms were picked up from the algae lawns. To remove adhesions on the surface, 5 mature or 10 immature animals taken from the culture medium were carefully cleaned with sterile Guillard's F/2 medium before their DNA was extracted. The animals were starved for 5 days in the starvation group. These samples together with 4 weeks of culturing algae and its living substrates were prepared for comparing the natural microbiomes of "mature worms". Total DNA extractions were followed by the instructions of the DNeasy ®Blood &Tissue kits (Qiagen, Hilden, Germany). DNA was eluted with 50 µl of elution buffer and stored at −80 °C.

The 16S rRNA gene primer pair 27F/338R was used to test the quality of DNA samples in a 20 µl duplex PCR reaction with Phusion High Fidelity DNA Polymerase (Thermo Fisher Scientific, Waltham, USA). The following PCR program was followed: a 3 min pre-denaturation step at 98 °C, followed by 35 cycles (98 °C for 10 s, 58 °C for 30 s, and 72 °C for 1 min), and thereafter a final 5 min elongation step at 72 °C. The PCR products were checked with 2% agarose gel, and the DNA samples having clear positive bands of the correct size were chosen for sequencing of the bacterial 16 S V1-V2 region on the MiSeq platform (Illumina) at the Institute for Clinical Molecular Biology, Kiel University. PCRs including both positive and negative controls were carried out under sterile conditions. For the negative control, sterile deionized water was used instead of template DNA.

The required materials and clean bench were all sterilized using 70% ethanol or followed by 20 min UV irradiation prior to sample isolation. Besides the centrifugation step, all remaining DNA isolation steps were carried out under a clean bench to avoid contamination in the laboratory environment.

**Bacterial qPCR.** The total bacterial DNA load of worms was measured using real-time quantitative PCR, which is based on the amplification of conserved segments of the 16S rRNA genes. The reaction component including 5 µl of 2x qPCR BIOSyGreen Mix Hi-ROX (London, UK), 0.5 µl of sense primer (5 µM), 0.5 µl of anti-sense primer (5 µM), 1 µl of template DNA (2 ng/µl) and 3 µl of HPLC $H_2O$. The amplification program was as follows: 1 cycle (95 °C, 10 min), 40 cycles (95 °C, 15 s; 60 °C, 20 s; 17 °C, 35 s). The whole reaction was performed using the StepOnePlus™. The same DNA samples were used for Miseq sequencing and qPCR amplification. The primer set for 16 S was as follows: 8FM (5′-AGAGTTTGATCMTGGCTCAG-3′); Bact515R (5′-TTACCGCGGCKGCTGGCAC-3′)[54]. In this case, the ribosomal protein L12 (rpl12) of *M. lignano* served as a reference gene. The primer pair was: 5′-GACAAGGTTAAC-GACGGCTC-3′; 5′-TATAGCAGCCGGTGTGTCAA-3′[77].

**Circadian rhythmicity of the microbiota of M. lignano.** Experiments were set up in a controlled incubator room at 20 °C with 4 µmol m$^{-2}$ S$^{-1}$ luminous intensity, 66% RLF humidity, and 12 h/12 h day/night rhythms. To avoid the discrepancy generated from the samples, all animals were 4-week-old adults with 1-week constant adaption. The study rhythm was carried out on a 24 h scale, 10 animals per group with 5 replicates were used. They were raised under identical conditions and picked up every 3 h. The corresponding substrate samples were simultaneously picked up with the worms. To decrease the risk of contamination, animal culturing always was done using sterilized petri dishes within germ-free plastic chambers. All handling procedures were performed under the clean bench.

**Analysis of bacterial communities.** Raw reads were first filtered by fastp using parameters -M 20 -q 20 to discard low-quality sequences[78]. To harness the full potential of deep-sequenced high-quality reads and to gain insight into the strain-level host-associated microbiome diversity, amplicon sequence variants (ASVs) were resolved from error-corrected biological sequences down to the level of single-nucleotide differences, instead of clustering reads into operational taxonomic units (OTUs) on a 97% similarity[79]. Error prediction, feature counting, and chimera detection and removal were performed using the Dada2 pipeline[80]. Taxonomy was assigned to each ASV referring to a Dada2-formatted training dataset[80] derived from the Silva v132 release[81]. Phylogenetic reconstruction was performed using the phangorn package[82]. Further bacterial community analysis was conducted using R packages[83] Phyloseq[84] and DEseq2[85] in RStudio[86]. To compensate for the uneven sequencing effort across different samples, ASV counts in individual samples were normalized to the median value of total counts in the analyzed groups and reported as either normalized counts or percentage proportions. A-diversity measurements were estimated using QIIME 2[87] after samples were rarefied to 23152 sequences per sample.

**Statistical analysis.** In all experiments, statistical analysis was performed with Prism version 7.0. The unpaired two-tailed Student's *t*-test was used for the analysis of differences between the two groups. One-way ANOVA followed by a Turkey posthoc test was used for the multiple comparisons. Two-way ANOVA was performed to analyze two different categorical independent variables on one continuous dependent variable. All values of histograms were means ± SD. The survival curves of the lifespan were analyzed with the log-rank (Mantel-Cox) test. The levels of significant difference were defined as *$p < 0.05$, **$p < 0.01$, ***$p < 0.001$, and ****$p < 0.0001$. The cycling statistical p-value assignments used JTK non-parametric test[88].

**Reporting summary.** Further information on research design is available in the Nature Portfolio Reporting Summary linked to this article.

## Data availability

The authors declare that all data supporting the findings of this study are available in the manuscript. The 16S amplicon sequencing data generated in this study are available in the NCBI BioProject database under the accession number PRJNA933008. Other relevant data are available from the corresponding author upon request.

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

## Acknowledgements

We would like to thank Britta Laubenstein, Christiane Sandberg, Heidrun Ließegang, and Katja Cloppenborg-Schmidt for excellent technical assistance. The work was funded by the German Science Foundation DFG (CRC1182, Projects C2, C1, A4, Z2, Z3, and INST 257/591-1 FUGG) and the Chinese Scholarship Council (CRC).

## Author contributions

Y.M., I.B., J.F.B., U.B., and T.R. designed the study and planned the experimental design, and Y.M., J.H., M.S., and J.v.F. performed the experiments. All authors contributed to writing or editing the manuscript.

## Funding

## Competing interests

The authors declare no competing interests.
