## [Peer Review File · Communications Biology]

Reviewers' comments:

Reviewer #1 (Remarks to the Author):

Ma and colleagues characterized the microbiota of the flatworm *Macrostomum lignano* and demonstrated that different developmental stages of the animal have distinct microbiota composition, there is circadian rhythm in microbiota, and that microbiota increases fitness of the animals when the food supply is limited.

This is a descriptive study but it is the first description of microbiota of *M. lignano*, and as such it adds a new layer of knowledge about this model organism and sets the stage to use it as a model to investigate host-microbiota dynamics.

The work is of sufficient quality and the conclusions are well-supported by the provided experimental data. I don't have major criticisms about the work but have several suggestions to clarify and improve some points:

Line 62: clarify what advantages are imparted by microbiota.

Line 66: repetitive sentence.

Line 143: define SWM.

Line 225: triangle, not rectangle

Line 313-314: add discussion about advantage for regeneration, if any. Additional experimental data on how regeneration is affected by microbiota would be ideal, but not essential for this work.

Line 349: Any clues if the enriched or depleted taxa can give metabolic advantages? Or other ideas why these taxa would be more specific?

Line 400: which line of *M. lignano* was used? I assume University of Innsbruck have provided the name of the line, since there are several different lines currently in use around the world.

Line 425: were the animals kept in petri dishes or multiwell plates?

Line 429: to slower -> to slow? Not sure I understand how this slows feces decay, please explain.

Reviewer #2 (Remarks to the Author):

--This information is also provided in an attachment--

Ma and colleagues present a lengthy study on several elements of the microbiome associated with the marine flatworm *Macrostomum lignano*. Their data suggest that this microbiome is taxonomically and quantitatively dynamic in response to development and ecological factors. These findings are widely consistent with other animal systems in the sea as well as on land. They now claim that *M. lignano* "can serve as a general model for host-microbe interactions in marine invertebrates." This repeated statement is far from justified based on the data presented by the authors. Specifically, there is no clear significance to using *M. lignano* as an experimental model—as opposed to a myriad of other systems—and the data presented here do not justify its use. If there is, then the authors must restructure this entire manuscript to systemically show how *M. lignano* is comparable to other models of microbiome research. See these qualifications in Ruby (2008; *Nature Reviews Microbiology* 6: 752-762) and Douglas (2019; *Nature Reviews Microbiology* 17: 764-775). Outside of this, the surplus of data are incredibly difficult to follow, making it seem like the authors are presenting several data

types without any clear reason for why they are doing so and how they are connected. All of these concerns should be able to be met in a major revision and I look forward to seeing an improved version of this manuscript.

Minor

Introduction

General: Avoid phrases such as, "Very impressive examples" or "The most impressive examples" or "very successful efforts."

Ln 40-41: be consistent with bacteria vs microbiome/microorganisms, as the latter includes more than 'just' bacteria.

Ln 45: combine these references.

Ln 51-52: the phrase "There are intracellular as well as extracellular symbionts," does not needed here. If you insist on keeping it then find a more natural way to integrate it.

Ln 63: Start a new paragraph with "Despite the very successful,,"; this current one is far (!) too long.

Ln 63-67: this is both repetitive and isn't clear.

Ln 70-71: You haven't provided enough context about sponges, coral, and deep-sea invertebrates to say "Besides these highly interesting and well characterized..."

Ln 72: Much of the microbiota are neutral, so saying "specialized" is not justified. This would only apply to deep-sea invertebrates, as far as we know.

Ln 73: "more diverse" than what? And, give some estimation of what "diverse" is. See, for example, O'Brien et al. (2019; mBio).

Ln 76-77: But why? You could have chosen any species then provide background on its life-history and genomics. I'd advise you to lead with the unique component(s) of its biology that justifies why it is worth studying. Perhaps this may be accomplished by rearranging this paragraph.

Materials & Methods

Ln 93: How does the Guillard's F/2 medium itself affect the microbiota (i.e., without *Nitzschia curvilineata*)? I ask this specifically because when feeding zooplankton microalgae cultured in Guillard's, the medium must be removed because that affects host physiology as well as the microbial load. This, I believe, is missing from 1-5 below, but please correct me if I mis-understood.

Ln 102: 'complex' is not the proper term here. Use 'diverse' while also stating the range of ASVs per stage.

Ln 118: total or relative abundance? Amplicon cannot provide information on total abundance since it is a qualitative tool.

Ln 128-130: Were the same bacterial lineages in the egg also uncharacterized in the adults? How many ASVs and was any phylogenetic comparison done to identify them?

Ln 141: for microbiomes, Shannon diversity is a questionable diversity estimate. Alpha diversity is a two-dimensional measure of diversity. The first dimension concerns the number of unique 'characters' in the dataset, which can be assessed for microbiome-related amplicon datasets using total taxa/ASVs and Faith's phylogenetic diversity. The second dimension concerns the frequency distribution of those characters, which can be assessed for microbiome-related amplicon datasets using measures of evenness and dominance. Each dimensional should be individually calculated and presented, and I request the authors do so. Moreover, there are several estimates of alpha diversity (e.g., Shannon, Simpson, Chao1) that attempt to merge these two dimensions into one assessment and, in doing so, they do not fully represent the multi-dimensionality of alpha diversity. If you wish to keep Shannon diversity, then please also present total taxa / Faith's phylogenetic diversity and dominance / evenness.

Ln 145: is the UniFrac measure consistent with this statement? This measure has more information (i.e., phylogeny, presence, relative abundance) and is more informative for diverse ecological communities than B-C (i.e., presence, relative abundance).

Ln 163: testing the neutral model for "enriched or depleted" is not the correct analysis, as this should correspond with some differential abundance test. The text below this has the correct phrasing, but the term "enriched or depleted" is feels off.

Ln 174: what does "compatible" mean here? I believe that the microbes above this neural expectation would be considered non-random symbionts.

Ln 229-230: without a systematic assessment of development, this sentence cannot be accurate. Please be more specific than 'development.' Do you mean developmental pace? There could be major developmental defects that cannot be seen through this course assessment.

Ln 232: please change to "as can be defined." There are other measures of fitness.

Ln 238: was this 35% significant? If so, please state.

Ln 260: Please divide this into 2 or 3 appropriately sized and topically consistent paragraphs.

Discussion

General: this under references the literature and does not properly put this system into context for comparison.

Ln 307: if you fall between these two extremes, then doesn't that make this system normal? Why would it be a candidate to be a model species?

Ln 314: there are more ways to know the significance of a symbiosis than gnotobiotic comparisons.

Ln 344: Please provide examples from all major animal lineages.

Ln 350: I am not sure this is particularly accurate.

Methods

Ln 425: fifty should not be capitalized.

Ln 432: why were these data not discussed in the results?

Figure 1

A: what is the label for the y-axis?

B & C: It may be clear to either present the raw points or one per with average and standard deviation. The combination of dots, lines, and boxes is confusing. Please simplify.

In D, it seems very repetitive to have the legend, the labels, as well as the images all on/next to the graph. The legend should suffice. Also, please consider a font size that is more legible once in print.

Caption: Ln 108, spell out genus since figures are stand-alone objects; Ln 110, there is a strange character that should be beta.

Figure 2

A & B: For consistency, I think that would be more appropriate to use 100 on the x-axis than 1?

Figure 3

B: How does microbial abundance change with development? I ask because this is a key part of the manuscript.

Figure 4

A: What are the vertical arrows for? Please align the text (i.e., first through third transfer) to the right.

D: How is there significance between 'food sufficient' individuals? Those bars and their deviation overlap completely. This has the same 'degree of significance' (i.e., ***) as the data in 'E,' which should major visual differences.

Legend: please correct ")))"

Figure 5

This is visually challenging, and I request several vital changes. First, the horizontal spacing needs improvement, as the circles for PCM replicates overlap significantly. Second, many of the replicates are visually identical and thus are excessive, so please simplify by presenting averages. Third, without any additional phylogenetic information, having several ASVs with the same name (e.g., Alteromonadales) makes it impossible to understand these microbes. Perhaps it would be worth

collapsing the phylogeny by these major groups. Forth, the scaling here is impossible. How can one know the difference between 0.03 and 0.007 by eye? These data cannot possibly be specific and trustable to the 7th (!) decimal point. Fifth, which is light and which is dark periods? This information is key for circadian studies.

Figure 6

A: Please change this to Zeitgeber Time, as this is the official time measure in circadian biology.

B: "Alpha diversity measure" as a label on the y-axis is inappropriate.

A & B: the time standardization across this is inconsistent. This must be corrected.

General: Also, please add a shaded box across each component of the figure to represent light and dark periods. This is standard for circadian biology.

Legend: Please mention "C" here.

Dear Reviewers,

We have changed the manuscript according to your suggestions. Therefore, we modified figures 1, 2, 4, 5, and 6 and adjusted the results section accordingly. Moreover, we modified the Introduction and Discussion to focus more on the most relevant aspects of the manuscript. Below you will find a point-to-point reply to your suggestions.

Reviewer #1 (Remarks to the Author):

Ma and colleagues characterized the microbiota of the flatworm *Macrostomum lignano* and demonstrated that different developmental stages of the animal have distinct microbiota composition, there is circadian rhythm in microbiota, and that microbiota increases fitness of the animals when the food supply is limited.

This is a descriptive study but it is the first description of microbiota of *M. lignano*, and as such it adds a new layer of knowledge about this model organism and sets the stage to use it as a model to investigate host-microbiota dynamics.

The work is of sufficient quality and the conclusions are well-supported by the provided experimental data. I don't have major criticisms about the work but have several suggestions to clarify and improve some points:

Line 62: clarify what advantages are imparted by microbiota.

Line 61-76: Modified. Here we have added a more elaborated description of the advantages of having a microbiota.

Line 66: repetitive sentence.

Line 80: We have modified this. Redundant expressions have been removed.

Line 143: define SWM.

We have added this (line 172).

Line 225: triangle, not rectangle

Triangle replaced rectangle.

Line 313-314: add discussion about advantage for regeneration, if any. Additional experimental data on how regeneration is affected by microbiota would be ideal, but not essential for this work.

We now added a part to the discussion section where this aspect of regeneration is now highlighted (lines 392-403).

Line 349: Any clues if the enriched or depleted taxa can give metabolic advantages? Or other ideas why these taxa would be more specific?

We included a short discussion about some of the most relevant microbes focusing on their potential role (lines 413-423).

Line 400: which line of *M. lignano* was used? I assume University of Innsbruck have provided the name of the line, since there are several different lines currently in use around the world.

The DV1 inbred *M. lignano* line used in this study was described previously (line 467).

Line 425: were the animals kept in petri dishes or multiwell plates?

Line 451, 471: According to the different situations, the animals were kept both in Petri dishes and multiwell plates in our lab. For daily preservation and numbers in high demand (above 50), we use Petri dishes filled with 20ml artificial seawater. For the small number of individuals required for the multiple replicates, we use multiwell plates.

Line 429: to slower -> to slow? Not sure I understand how this slows feces decay, please explain.

Has been changed. We also included a sentence that explains the rationale behind this.

Reviewer #2 (Remarks to the Author):

--This information is also provided in an attachment--

Ma and colleagues present a lengthy study on several elements of the microbiome associated with the marine flatworm *Macrostomum lignano*. Their data suggest that this microbiome is taxonomically and quantitatively dynamic in response to development and ecological factors. These findings are widely consistent with other animal systems in the sea as well as on land. They now claim that *M. lignano* “can serve as a general model for host-microbe interactions in marine invertebrates.” This repeated statement is far from justified based on the data presented by the authors. Specifically, there is no clear significance to using *M. lignano* as an experimental model—as opposed to a myriad of other systems—and the data presented here do not justify its use. If there is, then the authors must restructure this entire manuscript to systemically show how *M. lignano* is comparable to other models of microbiome research. See these qualifications in Ruby (2008; *Nature Reviews Microbiology* 6: 752-762) and Douglas (2019; *Nature Reviews Microbiology* 17: 764-775). Outside of this, the surplus of data are incredibly difficult to follow, making it seem like the authors are presenting several data types without any clear reason for why they are doing so and how they are connected. All of these concerns should be able to be met in a major revision and I look forward to seeing an improved version of this manuscript.

Minor

Introduction

General: Avoid phrases such as, “Very impressive examples” or “The most impressive examples” or “very successful efforts.”

This has been changed.

Ln 40-41: be consistent with bacteria vs microbiome/microorganisms, as the latter includes more than ‘just’ bacteria.

This has been changed.

Ln 45: combine these references.

Have been combined.

Ln 51-52: the phrase “There are intracellular as well as extracellular symbionts,” does not needed here. If you insist on keeping it then find a more natural way to integrate it.

Replaced by “Understanding this complex interplay between microbiota and host, and the effects of an out-of-balance interaction, requires appropriate animal models.”

Ln 63: Start a new paragraph with “Despite the very successful,”; this current one is far (!) too long.

Introduction is now more structured.

Ln 63-67: this is both repetitive and isn’t clear.

This has been replaced.

Ln 70-71: You haven’t provided enough context about sponges, coral, and deep-sea invertebrates to say “Besides these highly interesting and well characterized...”

We have added a paragraph to the introduction to elaborate more on these models (lines 77-85).

Ln 72: Much of the microbiota are neutral, so saying “specialized” is not justified. This would only apply to deep-sea invertebrates, as far as we know.

This has been replaced.

Ln 73: “more diverse” than what? And, give some estimation of what “diverse” is. See, for example, O’Brien et al. (2019; mBio).

This has been changed and the criteria of O’Brien et al have been included (lines 74-76).

Ln 76-77: But why? You could have chosen any species then provide background on its life-history and genomics. I’d advise you to lead with the unique component(s) of its biology that justifies why it is worth studying. Perhaps this may be accomplished by rearranging this paragraph.

We changed the paragraph and explained the motivation behind this in detail (lines 86-95).

Materials & Methods

Ln 93: How does the Guillard’s F/2 medium itself affect the microbiota (i.e., without *Nitzschia curvilineata*)? I ask this specifically because when feeding zooplankton microalgae cultured in Guillard’s, the medium must be removed because that affects host physiology as well as the microbial load. This, I believe, is missing from 1-5 below, but please correct me if I mis-understood.

We can not measure the effects of Guillard’s medium, as all experiments were done in this medium. With respect to the effects of *Nitzschia curvilineata* presence, we present these results as part of Figure 3.

Ln 102: ‘complex’ is not the proper term here. Use ‘diverse’ while also stating the range of ASVs per stage.

Has been changed and the ASVs per stage were added (lines 132-136).

Ln 118: total or relative abundance? Amplicon cannot provide information on total abundance since it is a qualitative tool.

This has been changed (line 120).

Ln 128-130: Were the same bacterial lineages in the egg also uncharacterized in the adults? How many ASVs and was any phylogenetic comparison done to identify them?

The bacterial lineages in the egg also characterized in the adults. According to the family taxonomic order, we found four undefined species. They belong to four orders, namely Rhizobiales, Rhodobacterales, Caulobacterales, Enterobacteriales. The first three orders are closely related and belong to the same class, namely Alphaproteobacteria. While the last group, Enterobacteriales belong to the Gammaproteobacteria.

Ln141: for microbiomes, Shannon diversity is a questionable diversity estimate. Alpha diversity is a two-dimensional measure of diversity. The first dimension concerns the number of unique ‘characters’ in the dataset, which can be assessed for microbiome-related amplicon datasets using total taxa/ASVs and Faith’s phylogenetic diversity. The second dimension concerns the frequency distribution of those characters, which can be assessed for microbiome-related amplicon datasets using measures of evenness and dominance. Each dimensional should be individually calculated and presented, and I request the authors do so. Moreover, there are several estimates of alpha diversity (e.g., Shannon, Simpson, Chao1) that attempt to merge

these two dimensions into one assessment and, in doing so, they do not fully represent the multi-dimensionality of alpha diversity. If you wish to keep Shannon diversity, then please also present total taxa / Faith's phylogenetic diversity and dominance / evenness.

Thank you very much for your detailed explanation. Here, we changed Figure 1 accordingly and included the relevant measures of the alpha-diversity (Fig. 1B-E). This is also described in detail in the results section.

Ln 145: is the UniFrac measure consistent with this statement? This measure has more information (i.e., phylogeny, presence, relative abundance) and is more informative for diverse ecological communities than B-C (i.e., presence, relative abundance).

We agree that the Unifrac measurements are more informative but presenting all the results for the five sets of data in one graph inevitably complicates the presentation of the results. We put it in the attachment, and here it is more clearly expressed. A non-metric multidimensional scaling (NMDS) plot can show separated group differences, while a bar graph is more intuitive to visualize relative abundance of microbiome.

Ln 163: testing the neutral model for "enriched or depleted" is not the correct analysis, as this should correspond with some differential abundance test. The text below this has the correct phrasing, but the term "enriched or depleted" is feels off.

We have corrected our expression and replaced it with the expressions of being present in high abundance in a small number of samples or being found in many samples in small amounts (outliers to the right and left, respectively).

Ln 174: what does "compatible" mean here? I believe that the microbes above this neural expectation would be considered non-random symbionts.

We have made changes with more specific expressions.

Ln 229-230: without a systematic assessment of development, this sentence cannot be accurate. Please be more specific than 'development.' Do you mean developmental pace? There could be major developmental defects that cannot be seen through this course assessment.

We fully accept and have revised the word 'development' according to your comments.

Ln 232: please change to "as can be defined." There are other measures of fitness.

Modified, replacement of "as can be defined by the production of offspring," with "meaning offspring production".

Ln 238: was this 35% significant? If so, please state.

Yes, we added p-value.

Ln 260: Please divide this into 2 or 3 appropriately sized and topically consistent paragraphs.

We have divided this accordingly.

Discussion

General: this under references the literature and does not properly put this system into context for comparison.

We restructured the discussion section and added literature to put the relevant points into context.

Ln 307: if you fall between these two extremes, then doesn't that make this system normal? Why would it be a candidate to be a model species?

We added an explanation to the manuscript (333-349).

Ln 314: there are more ways to know the significance of a symbiosis than gnotobiotic comparisons.

We removed the sentence 'The importance of a microbiota for its hosts can -be assessed by using gnotobiotic animals (MARQUES *et al.* 2006; DOUGLAS 2019)' with 'Here, using gnotobiotic animals is a straightforward approach, as it allows direct attribution of the observed phenotypes to the presence or absence of a microbiota'

Ln 344: Please provide examples from all major animal lineages.

Ln 416: Accept. "Diurnal rhythms of the microbiota of few invertebrates have also been shown, e.g. in the sea anemone *Nematostella vectensis*²², but the underlying mechanisms, as well as the physiological consequences, remain to be elucidated." ; the major animal examples were also presented (lines 61-76).

Ln 350: I am not sure this is particularly accurate.

Changed: "A first result of this study was that all host (*Macrostomum lignano*) associated samples show a profile that is uniquely distinguishable from the environmental samples."

Methods

Ln 425: fifty should not be capitalized.

We have modified "Fifty" to "fifty".

Ln 432: why were these data not discussed in the results?

We have added a description of these data. "We observed that the intestinal tract of the worms is slightly acidic ($5.2 \leq \text{pH} < 6.8$), while the sea water is neutral to alkaline ($7.4 < \text{pH} \leq 9.0$; Additional Fig. 4). A pH dependency of the microbial composition has also been shown in other systems {Firrman, 2022 #469}. "

Figure 1

A: what is the label for the y-axis?

Y-axis, presenting the bacteria proportion of different bacteria.

B & C: It may be clear to either present the raw points or one per with average and standard deviation. The combination of dots, lines, and boxes is confusing. Please simplify.

We now show the average deviation in the picture.

In D, it seems very repetitive to have the legend, the labels, as well as the images all on/next to the graph. The legend should suffice. Also, please consider a font size that is more legible once in print.

We deleted all repetitive legends and labels.

Caption: Ln 108, spell out genus since figures are stand-alone objects; Ln 110, there is a strange character that should be beta.

“Colored rectangles are used to distinguish different families of microorganisms.”; Character was replaced with “beta”.

Figure 2

A & B: For consistency, I think that would be more appropriate to use 100 on the x-axis than 1?

We have modified Figure 2 accordingly.

Figure 3

B: How does microbial abundance change with development? I ask because this is a key part of the manuscript.

Yes, this is an important part of the manuscript. This part of the results are shown in Figure1. Figure 3 only shows the mature worms (MWs), one is the fed group and the other is the starved group.

Figure 4

A: What are the vertical arrows for? Please align the text (i.e., first through third transfer) to the right.

We added an explanation of the vertical arrows. “Vertical arrows are used to indicate the worm transfer process.” And the text is right aligned. Please see the image below:

D: How is there significance between 'food sufficient' individuals? Those bars and their deviation overlap completely. This has the same 'degree of significance' (i.e., ***) as the data in 'E,' which should major visual differences.

We have changed this accordingly.

Legend: please correct “))”

Duplicates have been removed.

Figure 5

This is visually challenging, and I request several vital changes. First, the horizontal spacing needs improvement, as the circles for PCM replicates overlap significantly. Second, many of the replicates are visually identical and thus are excessive, so please simplify by presenting averages. Third, without any additional phylogenetic information, having several ASVs with the same name (e.g., Alteromonadales) makes it impossible to understand these microbes. Perhaps it would be worth

collapsing the phylogeny by these major groups. Forth, the scaling here is impossible. How can one know the difference between 0.03 and 0.007 by eye? These data cannot possibly be specific and trustable to the 7th (!) decimal point. Fifth, which is light and which is dark periods? This information is key for circadian studies.

We made the following modifications to Figure 5. In response to the first suggestion, the overlap between the symbols does not affect the identification of replicates, and this reduces the horizontal span, which is convenient for viewing the data of 8 sampling times and 16 groups. For the second suggestion, some replicates do have approximately the same abundance, but some groups have large differences, and some even have no differences. Simply, showing the average will obviously hide this part of the information. The third proposal has been modified as requested to provide additional phylogenetic information, explaining the same genus name in the text. For the fourth proposal, the scale of the ruler is modified to make it easier to distinguish. The last proposal, the light and dark periods are marked in the graph. Moreover, we changed the time to the Zeitgeber time.

Figure 6

A: Please change this to Zeitgeber Time, as this is the official time measure in circadian biology.

This has been changed for figures 5 and 6.

B: "Alpha diversity measure" as a label on the y-axis is inappropriate.

"Alpha diversity measure" was replaced by "Shannon index".

A & B: the time standardization across this is inconsistent. This must be corrected.

As in the question above, we uniformly adjust the time to Zeitgeber time. The picture shows an extra cycle in order to show the periodicity.

General: Also, please add a shaded box across each component of the figure to represent light and dark periods. This is standard for circadian biology.

Yes, we have added shaded boxes in the figure to present light and dark periods.

Legend: Please mention "C" here.

Ln271 We added "C".

REVIEWERS' COMMENTS:

Reviewer #2 (Remarks to the Author):

A very detailed revision. I look forward to seeing this in press.